# Assessing Boundary Condition and Parametric Uncertainty in Numerical-Weather-Prediction-Modeled, Long-Term Offshore Wind Speed Through Machine Learning and Analog Ensemble

Nicola Bodini [1], Weiming Hu [2], Mike Optis [1], Guido Cervone [2,3], and Stefano Alessandrini [3]

[1]National Renewable Energy Laboratory, Golden, Colorado, USA
[2]Department of Geography, Pennsylvania State University, University Park, PA, 16802, USA
[3]National Center for Atmospheric Research, Boulder, CO, USA

**Correspondence:** Nicola Bodini (nicola.bodini@nrel.gov)

**Abstract.** To accurately plan and manage wind power plants, not only does the time-varying wind resource at the site of interest need to be assessed, but also the uncertainty connected to this estimate. Numerical weather prediction (NWP) models at the mesoscale represent a valuable way to characterize the wind resource offshore, given the challenges connected with measuring hub height wind speed. The boundary condition and parametric uncertainty associated with modeled wind speed is often estimated by running a model ensemble. However, creating an NWP ensemble of long-term wind resource data over a large region represents a computational challenge. Here, we propose two approaches to temporally extrapolate wind speed boundary condition and parametric uncertainty using a more convenient setup where a mesoscale ensemble is run over a short-term period (1 year), and only a single model covers the desired long-term period (20 year). We quantify hub-height wind speed boundary condition and parametric uncertainty from the short-term model ensemble as its normalized across-ensemble standard deviation. Then, we develop and apply a gradient-boosting model and an analog ensemble approach to temporally extrapolate such uncertainty to the full 20-year period, where only a single model run is available. As a test case, we consider offshore wind resource characterization in the California Outer Continental Shelf. Both the proposed approaches provide accurate estimates of the long-term wind speed boundary condition and parametric uncertainty across the region ($R^2 > 0.75$), with the gradient-boosting model slightly outperforming the analog ensemble in terms of bias and centered root-mean-square error. At the three offshore wind energy lease areas in the region, we find a long-term median hourly uncertainty between 10% and 14% of the mean hub-height wind speed values. Finally, we assess the physical variability of the uncertainty estimates. In general, we find that the wind speed uncertainty increases closer to land. Also, neutral conditions have smaller uncertainty than the stable and unstable cases, and the modeled wind speed in winter has less boundary condition and parametric sensitivity than summer.

## 1   Introduction

Offshore wind energy keeps increasing its market penetration as an inexpensive and clean source of energy. In some areas of the world, such as the North Sea in Europe, offshore wind represents a well-established source of electricity, with a total installed capacity of about 15 GW, and a planned increase of up to 74 GW by 2030 (van Hoof, 2017). As the cost of offshore wind energy has been decreasing faster than expected (Stiesdal, 2016; Brandily, Tifenn, 2020), many other regions are currently planning to adopt offshore wind energy solutions to meet their energy needs. The United States fall within this group, with its offshore technical resource potential being estimated to be about twice the present national energy demand (Musial et al., 2016). While one single 30-MW offshore wind power plant has been operating since 2016 (Deepwater Wind, 2016), many other offshore wind plants are being planned, mostly concentrated along the Eastern Seaboard and the Outer Continental Shelf (OCS) off the coast of California, for a total of about 86 GW installed capacity expected by 2050 (Bureau of Ocean Energy Management, 2018).

Such extensive growth requires an accurate long-term characterization of the offshore wind resource (Brower, 2012). Direct observations of the wind resource offshore are oftentimes limited to buoys, which offer measurements at very limited heights. Hub-height measurement of the wind resource offshore can be achieved with either offshore meteorological towers (e.g., Neumann et al. (2004); Fabre et al. (2014); Peña et al. (2014); Kirincich (2020)) or floating lidars (Carbon Trust Offshore Wind Accelerator, 2018; OceanTech Services/DNV GL, 2020). However, the often prohibitive costs connected to both these measurement solutions limit their availability to a handful of locations, despite recent efforts in leveraging their punctual hub-height measurements for wind speed vertical extrapolation over a larger region (Bodini and Optis, 2020; Optis et al., 2021a). Given these constraints, numerical weather prediction (NWP) models at the mesoscale are often used to obtain an in space and time continuous mapping of the available offshore wind resource at the heights relevant for commercial wind power plant deployment (e.g., Mattar and Borvarán (2016); Salvação and Soares (2018)), with some studies (Papanastasiou et al., 2010; Steele et al., 2013; Arrillaga et al., 2016) also focusing on the validation of modeled coastal wind effects, such as sea breezes, which have a significant impact on offshore wind energy production (Archer et al., 2014).

Tens of billions of dollars will be invested in the U.S. offshore wind energy industry in the coming years. In order to minimize the financial risk associated with such major investments, not only is a characterization of the time-varying offshore wind resource needed, but an assessment of the uncertainty connected to this numerical prediction is of primary importance. A 1% uncertainty change in the mean wind resource translates to a 1.6%–1.8% uncertainty for the long-term wind plant annual energy production (Johnson et al., 2008; White, 2008; Holstag, 2013; Truepower, 2014), with a significant increase in the interest rates for new project financing. However, assessing the uncertainty in modeled wind speed is a problematic task. NWP model ensembles tend to lead to an underdispersive behavior (Buizza et al., 2008; Alessandrini et al., 2013), so that

only a limited component of the actual wind speed error with respect to observations can be quantified. The full uncertainty in NWP-model-predicted wind speed can be quantified only when direct observations of the wind resource are available. In this scenario, the residuals between modeled and observed wind speed can be calculated, and the model error is quantified in terms of its bias (i.e., the mean of the residuals) and uncertainty (i.e., the standard deviation of the residuals). The obtained model uncertainty would then be added to the inherent uncertainty of the wind speed measurements by using a sum of squares approach (JCGM 100:2008, 2008). However, as we have already mentioned, direct observations of the wind resource are not always readily available, especially offshore, so that other ways to quantify at least specific components of the full wind speed uncertainty are needed.

When considering NWP models, the choices of the model setup and inputs have a direct impact on the model wind speed prediction, and therefore on its uncertainty. Hahmann et al. (2020) recently provided a detailed analysis of the sensitivity in wind speed predicted by NWP models, as part of the development of the New European Wind Atlas. Among the various sources of uncertainty, the choices of the planetary boundary layer (PBL) scheme (Ruiz et al., 2010; Carvalho et al., 2014a; Hahmann et al., 2015; Olsen et al., 2017) and of the large-scale atmospheric forcing (Carvalho et al., 2014b; Siuta et al., 2017) have been shown to have a major impact. Model resolution (Hahmann et al., 2015; Olsen et al., 2017), spin-up time (Hahmann et al., 2015) and data assimilation techniques (Ulazia et al., 2016) have also been shown to contribute to the wind speed sensitivity. The variability of modeled wind speed that derives from all the different model choices leads to what we will call boundary condition and parametric uncertainty of the modeled wind speed. Optis et al. (2021b) recently explored best practices for quantifying and communicating NWP-modeled wind speed boundary condition and parametric uncertainty offshore. In their approach, an ensemble of Weather Research and Forecasting (WRF) model (Skamarock et al., 2008) simulations is created by considering different WRF versions, namelists, and external forcings, and the wind speed boundary condition and parametric uncertainty is then quantified in terms of its across-ensemble variability. The use of numerical ensembles for uncertainty quantification is not exclusive to the wind energy community, as it has been extensively applied in ample spectrum fields (e.g., Zhu (2005); Parker (2013); Murphy et al. (2004)). However, running a NWP ensemble across a large region and for the long-term period needed for an accurate characterization of the naturally varying wind resource is computationally prohibitive, so that innovative and more computationally efficient ways are needed to quantify some components of the long-term wind speed uncertainty.

Here, we consider wind speed characterization in the California OCS, and we propose and compare two innovative techniques for modeled wind speed long-term boundary condition and parametric uncertainty quantification. To do so, we consider a setup that is computationally more affordable, wherein WRF ensembles are only run over a short period (1 year), and are accompanied by a single, long-term (20 years) WRF simulation. First, we use a machine-learning algorithm to temporally extrapolate the WRF-based boundary condition and parametric uncertainty from the ensemble year to the full 20-year period. While machine learning has been successfully applied to various atmospheric (e.g., Xingjian et al. (2015); Gentine et al. (2018); Bodini et al. (2020)) and wind-energy-related (e.g., Clifton et al. (2013); Arcos Jiménez et al. (2018); Optis and Perr-Sauer (2019)) problems, this represents, to the authors' knowledge, its first application for NWP uncertainty extrapolation. We compare the machine-learning-based approach with the predictions from the analog ensemble (AnEn) technique (Delle Monache

**Table 1.** Common specification for all of the WRF runs considered in the analysis.

| Feature | Specification |
|---|---|
| WRF version | 4.1.2 |
| Nesting | 6 km, 2 km |
| Vertical levels | 61 |
| Atmospheric nudging | Spectral nudging on a 6-km domain, applied every 6 hours |
| Microphysics | Ferrier |
| Longwave radiation | Rapid radiative transfer model |
| Shortwave radiation | Rapid radiative transfer model |
| Topographic database | Global multiresolution terrain elevation data from the United States Geological Service and National Geospatial-Intelligence Agency |
| Land-use data | Moderate resolution imaging spectroradiometer 30s |
| Cumulus parameterization | Kain-Fritsch |

et al., 2013) to quantify uncertainty in the wind resource from the variability in modeled cases with similar atmospheric conditions. Typical applications of AnEn include renewable energy probabilistic forecast, for both solar (Alessandrini et al., 2015a, b; Cervone et al., 2017) and wind (Junk et al., 2015; Vanvyve et al., 2015) energy. The use of AnEn for long-term offshore wind speed uncertainty quantification represents a novel application of the technique.

In the remainder of this paper, we describe the experimental setup and our proposed methods to quantify and temporally extrapolate modeled wind speed boundary condition and parametric uncertainty in Section 2. Section 3 validates the techniques used, and compares the mean long-term predictions from the two approaches. Also, we discuss physical insights into the main drivers for offshore wind speed boundary condition and parametric uncertainty. Finally, we conclude and suggest future work in Section 4.

## 2    Data and Methods

### 2.1    Numerical simulation setup

We consider a 20-year numerical data set recently developed by the National Renewable Energy Laboratory to provide accurate cost estimates for floating wind in the California OCS (Figure 1). As described in detail in Optis et al. (2020), this product includes a single WRF setup that is run for a 20-year period (2000-2019), and an additional 15 WRF ensemble members run over a single year (2017), which was selected because of strong data coverage from the network of buoy and coastal radar observations used for model validation. All of the simulations are run with the common attributes in Table 1. A total of over 200,000 grid cells are included in the WRF inner domain, which we consider in our uncertainty analysis. The 16 WRF ensemble

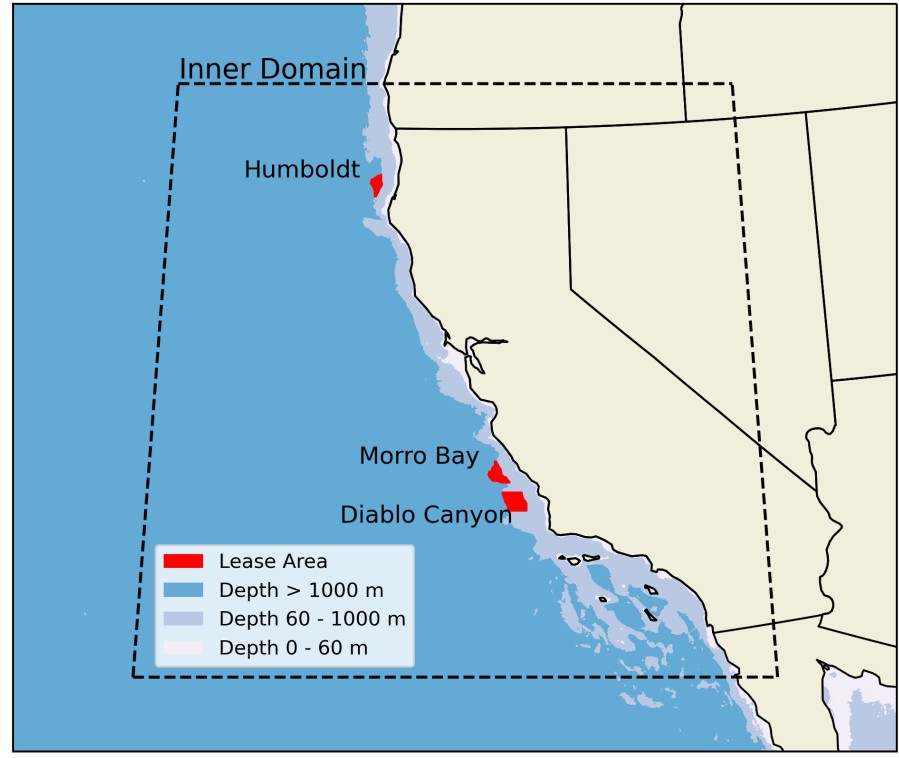

**Figure 1.** Map of the inner domain of the WRF numerical simulations for the California OCS. The current three wind energy lease areas are shown in red.

members are constructed based on variations of boundary conditions and key WRF model parameters that previous research determined to have a primary impact on modeled wind speed:

  – Reanalysis forcing product: selected between ERA5, developed by the European Centre for Medium-Range Weather Forecasts (Hersbach et al. (2020)), and the Modern-Era Retrospective analysis for Research and Applications, Version 2 ( Gelaro et al. (2017)), developed by the National Aeronautics and Space Administration.

  – PBL parameterization: chosen between the Mellor-Yamada-Nakanishi-Niino (MYNN, Nakanishi and Niino (2004)) and the Yonsei University (Hong et al. (2006)) schemes.

  – Sea surface temperature product: selected between the Operational Sea Surface Temperature and Sea Ice Analysis (OS-TIA) data set produced by the UK Met Office (Donlon et al., 2012) and the National Center for Environmental Prediction Real-Time Global product.

  – Land surface model: chosen between the Noah model and the updated Noah multiparameterization model (Niu et al., 2011).

The setup we chose to use for the single long-term WRF run is the result of a validation process. We compared and validated the 16 model setups with observations from buoys from the National Data Buoy Center and coastal radar measurements from the National Oceanic and Atmospheric Administration profiler network. While these are the only observations in the area available for model validation, these data sets cannot be used for quantifying the model uncertainty in wind speed. In fact, both data sources have significant limitations that do not allow for a direct comparison with offshore WRF data at heights relevant

for wind energy development. On one hand, buoys only measure wind speed close to the water level, which can have a very different regime than the hub-height winds. On the other hand, the coastal radars measure at more relevant heights for wind energy, but only at the interface between the ocean and land. Results from the validation (whose details can be found in Optis et al. (2020)) revealed that the WRF setup providing the most accurate results is the one using ERA5 as reanalysis product, MYNN as a PBL scheme, OSTIA as an SST product, and Noah as a land surface model. Therefore, we selected and adopted

this WRF setup for the single 20-year WRF run.

In our analysis, we use hourly average data (calculated from 5-minute WRF raw output), and we quantify the WRF wind speed boundary condition and parametric sensitivity in terms of the across-ensemble standard deviation of the WRF-predicted 100-m wind speed at any hour, $t$, normalized by the hourly average 100-m wind speed itself:

$$\sigma_{\mathrm{WS}}(t) = \frac{\frac{1}{N} \sum_{i=1}^{N} \left( \mathrm{WS}_i(t) - \overline{\mathrm{WS}}(t) \right)^2}{\mathrm{WS}_1(t)} \tag{1}$$

where $\mathrm{WS}_i$ is the mean hourly 100-m wind speed from each ensemble member, $\overline{\mathrm{WS}}$ is the mean hourly wind speed averaged across the 16 ensemble members, $\mathrm{WS}_1$ is the mean hourly 100-m wind speed from the WRF control run (i.e., the one used for the long-term period), and $N = 16$ is the total number of WRF ensemble members. Within a numerical ensemble framework, the use of (normalized) standard deviation as a primary uncertainty metric has been recommended by Optis et al. (2021b), as it provides more consistent estimates than the ensemble interquartile range. While we acknowledge that our quantification of the

WRF wind speed boundary condition and parametric sensitivity is going to be limited by the finite number of choices made to construct the ensemble members, we note how the considered set of settings represents either state-of-the-art products or the most popular and widely accepted choices for WRF applied to wind resource characterization. In the next sections, we present the two approaches we propose to temporally extrapolate this boundary condition and parametric uncertainty from 2017 (i.e., the only year where the uncertainty can be directly calculated from the WRF ensemble members) to the remaining 19 years.

## 2.2  Machine-learning approach

The first approach we use to temporally extrapolate the boundary condition and parametric uncertainty in 100-m modeled wind speed is a machine-learning gradient-boosting model (GBM) (Friedman, 2002). We select a GBM because ensemble-based algorithms are known to provide robust and accurate predictions in nonlinear problems. Moreover, we have tested a set of other machine-learning algorithms (random forest, generalized additive model), and the GBM provided the lowest prediction

error. With this approach, at each of the more than 200,000 grid cells, we train the model on calendar year 2017 to predict the WRF across-ensemble standard deviation of hourly average 100-m wind speed, normalized by the hourly average 100-m wind speed itself (Eq. 1). We then apply the trained model to quantify the modeled wind speed boundary condition and parametric

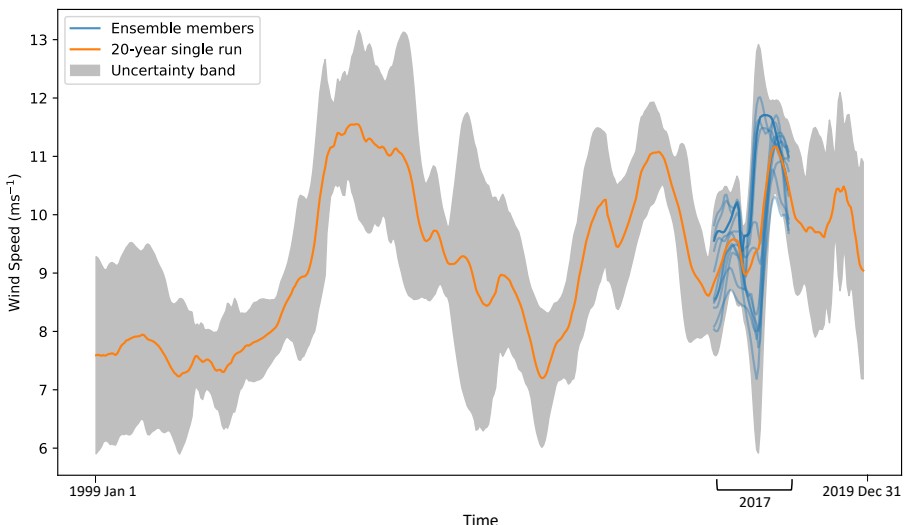

**Figure 2.** Qualitative illustration of the concept used to temporally extrapolate the 100-m modeled wind speed boundary condition and parametric uncertainty through the proposed machine-learning and analog ensemble approaches. Wind speed uncertainty is directly quantified as its WRF across-ensemble normalized standard deviation (Eq. 1) on the single year in which ensembles were run, and then extrapolated to the remaining 19 years, which were run with only a single WRF setup, using the proposed GBM and AnEn approaches.

uncertainty in the remaining 19 years where only the single WRF run is available (Figure 2). The input features we use to feed the GBM are all taken (as hourly averages) from the single WRF setup that is run for the full 20-year period, and are:

– Wind speed at 100 m above ground level (AGL)

– Sine and cosine[1] of wind direction at 100 m AGL

– Air temperature at 40 m AGL

– Wind shear coefficient calculated between 10 and 200 m AGL

– Inverse of Obukhov length at 2 m AGL

– 100-m wind speed standard deviation calculated from the preceding 2 hours

– 100-m wind speed standard deviation calculated from the preceding 6 hours

– Sine and cosine[1] of the hour of the day

– Sine and cosine[1] of the month.

[1]Sine and cosine are used to preserve the cyclical nature of this feature. Both are needed because each value of sine only (or cosine only) is linked to two different values of the cyclical feature.

**Table 2.** Hyperparameters considered for the gradient-boosting model.

| Hyperparameter | Meaning | Sampled Values |
|---|---|---|
| Number of estimators | Number of trees in the forest | 100–300 |
| Learning rate | Rate by which the contribution of each tree is shrunk | 0.05–1 |
| Maximum depth | Maximum depth of the tree | 4–10 |
| Maximum number of features | Number of features to consider when looking for the best split | 1–7 |
| Minimum number of samples to split | Minimum number of samples required to split an internal node | 2–20 |
| Minimum number of samples for a leaf | Minimum number of samples required to be at a leaf node | 1–20 |

The distribution of these variables is presented and discussed in Section 3.1. We acknowledge that correlation exists between some of the input features used. However, we found that including all the features produced the best model accuracy. Also, principal component analysis could be applied to reduce the number of features used, but it is beyond the scope of our analysis. We also acknowledge how different choices for the atmospheric stability parameter could be explored, potentially leading to a more accurate representation of stability at the heights of interest for wind energy development compared to the near-surface Obukhov length.

The learning algorithm is trained using the root-mean-square error (RMSE) as a performance metric to tune the algorithm weights. To avoid overfitting, we implement regularization during the training of the learning algorithm, using the hyperparameters and value ranges listed in Table 2. At each site, we sampled 20 combinations of hyperparameters using a randomized cross validation. More details about the validation of the results from the proposed approach are given in Section 3.1.

## 2.3 Analog ensemble approach

The second approach we use to quantify and extrapolate modeled wind speed boundary condition and parametric uncertainty is based on the AnEn approach. At each site and for each hour (hereafter referred to as the "target hour"), the AnEn considers a set of atmospheric variables, which are consistent between the AnEn and the machine-learning approach, in a 3-hour window centered on the considered time stamp. Then, the AnEn looks for analog atmospheric conditions at the considered site using data from the single long-term WRF setup for year 2017. More in detail, the multivariate atmospheric state within the considered time window is compared with the atmospheric conditions modeled by the long-term WRF setup in all of the 3-hour time windows in 2017. For each hour in 2017, the AnEn calculates a similarity metric, formally defined as a multivariate Euclidean

**Table 3.** Variability of the RMSE of two weight optimization schemes: site-specific optimization and general domain optimization.

| Location | Variability RMSE (m/s) | |
| --- | --- | --- |
| | Site-specific optimization | General domain optimization |
| Morro Bay | 0.5709 | 0.5723 |
| Diablo Canyon | 0.5523 | 0.5548 |
| Humboldt | 0.7240 | 0.7261 |

distance measure (Delle Monache et al., 2013):

$$\|F_t, A_{t'}\| = \sum_{i=1}^{N} \frac{\omega_i}{\sigma_i} \sqrt{\sum_{j=-1,0,1\text{h}} \left(F_{i,t+j} - A_{i,t'+j}\right)^2}, \tag{2}$$

where $F$ is the WRF-modeled atmospheric state at the search window centered at time, $t$ (where $t$ varies over the full 20-year period); $A$ is the WRF-modeled atmospheric state over a window centered at time, $t'$ (where $t'$ varies in 2017); $N$ is the number of atmospheric variables being considered to identify the analogs; $\omega_i$ is the predictor weight associated with the atmospheric variable, $i$; and $\sigma_i$ is the standard deviation of the atmospheric variable, $i$, calculated over the search period.

Once the similarity metric is calculated for all of the hours ($t'$) in 2017, the $m$ analog hours with the highest similarity are selected to form the analog ensemble. Finally, the WRF-modeled across-ensemble 100-m wind speed standard deviation for each analog hour is considered. The average of these $m$ values, normalized by the 100-m wind speed at that target hour from the single long-term WRF run, is then used as the AnEn extrapolated uncertainty to associate with the initial target hour. As previously mentioned, the AnEn approach is then repeated at each grid cell in the domain and target hour to generate a $m$-member ensemble forecast for the full long-term period.

The results from the AnEn approach are sensitive to the predictor weights, $\omega_i$, and the number of analog members, $m$ (Junk et al., 2015; Alessandrini et al., 2019). Therefore, the AnEn approach first needs to be trained to determine the optimal values of these parameters, which maximize the accuracy of the AnEn predictions. In doing so, we use RMSE between 2017 AnEn-predicted and WRF-predicted wind speed uncertainty as the score metric for the optimization process. Training AnEn at each grid cell over our large domain is a computationally challenging task (Hu et al., 2021a). Therefore, we explore whether the same number of analogs and a single combination of optimized weights can be used over the whole domain. First, we perform a site-specific weight optimization at three sites, one for each wind energy lease area in the California OCS (Figure 1). Then, we alternatively tune a single combination of weights for all three sites, and we refer to this second approach as the general domain optimization. We compare the RMSE values from the two approaches in Table 3. At each site, the RMSE from the general domain optimization is only slightly higher ($< 0.5\%$ increase) than that from the much more expensive site-specific optimization. Therefore, we select the optimal number of analogs ($m = 16$, notably the same as the number of WRF ensemble members) and weights resulting from the AnEn training at the three sites all together. The optimal weights are listed in Table 4.

**Table 4.** Optimal weights associated with each physical variable in assessing the closeness of the match metric to identify the analogs.

| Physical Variable | AnEn Weight |
|---|---|
| Wind speed at 100 m AGL | 0.2 |
| Wind direction at 100 m AGL | 0.1 |
| Temperature at 40 m AGL | 0.1 |
| Inverse of Obukhov length at 2 m AGL | 0 |
| Standard deviation of 100-m wind speed over preceding 6 hours | 0.2 |
| Standard deviation of 100-m wind speed over preceding 2 hours | 0.2 |
| Shear exponent calculated between 200 m and 10 m AGL | 0.2 |

## 3  Results

### 3.1  Validation of the proposed approaches

As a first step, we need to assess the accuracy and validity of our proposed approaches for the wind speed boundary condition
and parametric uncertainty extrapolation. As an initial validation step, we compare the distributions of the atmospheric variables
used as inputs to the machine-learning and AnEn algorithms for 2017 with what is found in the full 20-year period. In fact, in
order for both approaches to be accurate, it is essential that the considered atmospheric variables in 2017 (i.e., where the models
are trained) experience a range of variability representative of the full 20-year period (i.e., where the models are applied). By
qualitatively comparing the distributions of the seven atmospheric variables at one of the three wind energy lease areas (Figure
3), the variability found in 2017 appears similar to what is found in the long-term 20-year period. To quantitatively confirm this,
we apply a Levene's test (Levene, 1961) to assess the equality of the variances of the two samples (2017 vs. 20-year period)
for each atmospheric variable. We find that for all the seven variables, the null hypothesis of homogeneity of variance cannot
be rejected (with $p$-values <0.05), thus confirming that it is highly unlikely that the variability found in 2017 is significantly
different from the variability in the long term. Similar results are found at the two other wind energy lease areas (figures not
shown).

After proving that the basic assumptions of the proposed approaches are validated by the data, we need to test the accuracy of
their predictions. To do so, at each grid cell we quantify the mean bias, centered or unbiased root-mean-square error (cRMSE),
and coefficient of determination, $R^2$, between the machine-learning or AnEn predictions and the actual WRF ensemble vari-
ability. For the machine-learning approach, we calculate these error metrics over a testing set, obtained by training the learning
algorithm on 80% of the 2017 data, and then testing it on the remaining 20%. To minimize the effects of the autocorrelation
in the data, we select the testing set without shuffling the data. Also, to test the algorithm on the full seasonal variability of
the atmospheric variables, we create the test set by selecting a contiguous 20% of data for each month in 2017. We compare
these results with the three error metrics calculated when applying the AnEn approach, at each grid cell, for 2017. The maps
of the various error metrics for the two approaches are shown in Figure 4. In general, the maps show how both approaches are

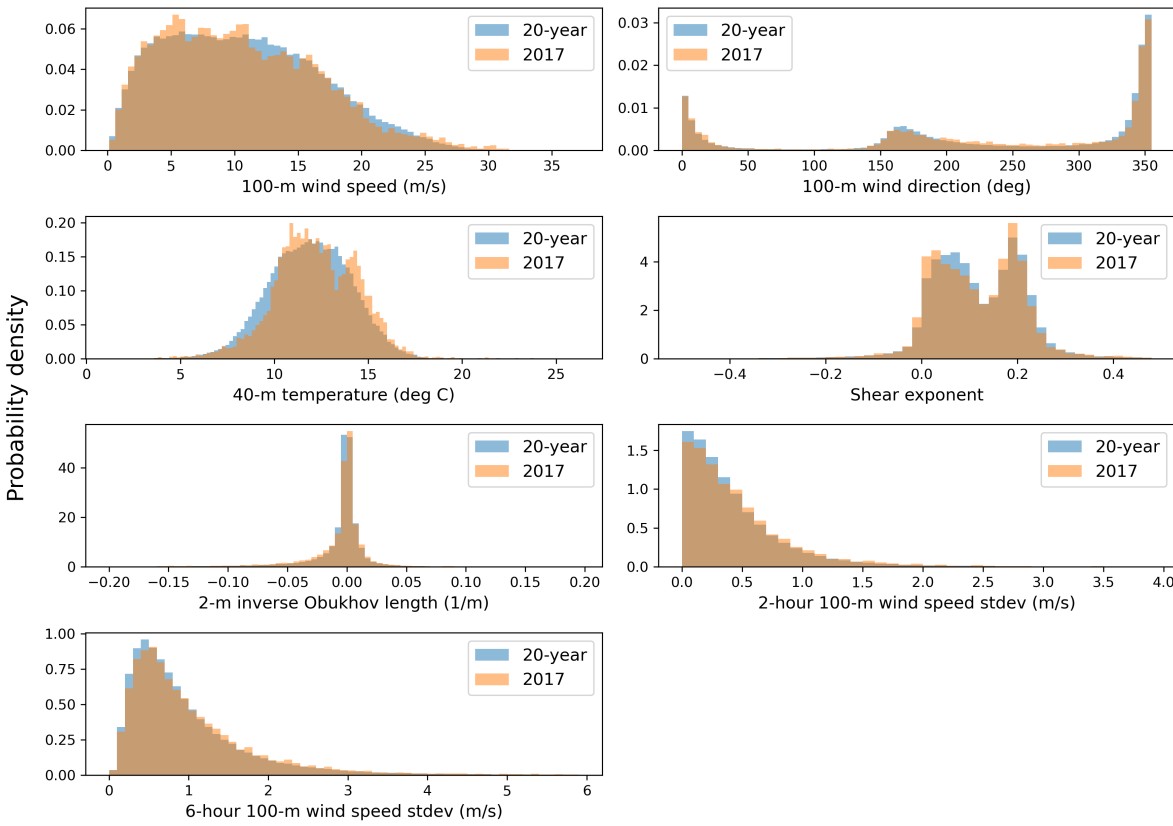

**Figure 3.** Distributions of the atmospheric variables considered as inputs to the machine learning and AnEn algorithms from 2017 only and from the full 20-year period for a single site within the Humboldt wind energy lease area. Data are expressed in terms of their probability densities.

capable of providing accurate predictions of the WRF boundary condition and parametric uncertainty across the whole off-shore domain. We find that the uncertainty predicted by the machine-learning model has a negligible bias (which is expressed as a percentage of the mean wind resource, the same as our normalized uncertainty metric) throughout the domain, whereas the uncertainty predictions from the AnEn approach are, on average, slightly lower than the WRF ensemble variability, with differences less than 3% at the three wind energy lease areas. The machine-learning approach also provides lower error after the bias is removed, especially closer to the coast, where the AnEn approach has local cRMSE values as high as 40%. On the other hand, we see that the AnEn approach provides a slightly stronger correspondence with the WRF data, with $R^2 > 0.80$ at the vast majority of the sites, whereas for the machine-learning model, $R^2 > 0.75$, with slightly lower values near the coast. The bias from the AnEn approach is likely because of the reduced length of the search period (1 year), which might be too limited for identifying a significant number (16) of analogs. This setup constrains the AnEn ability to account for rare events

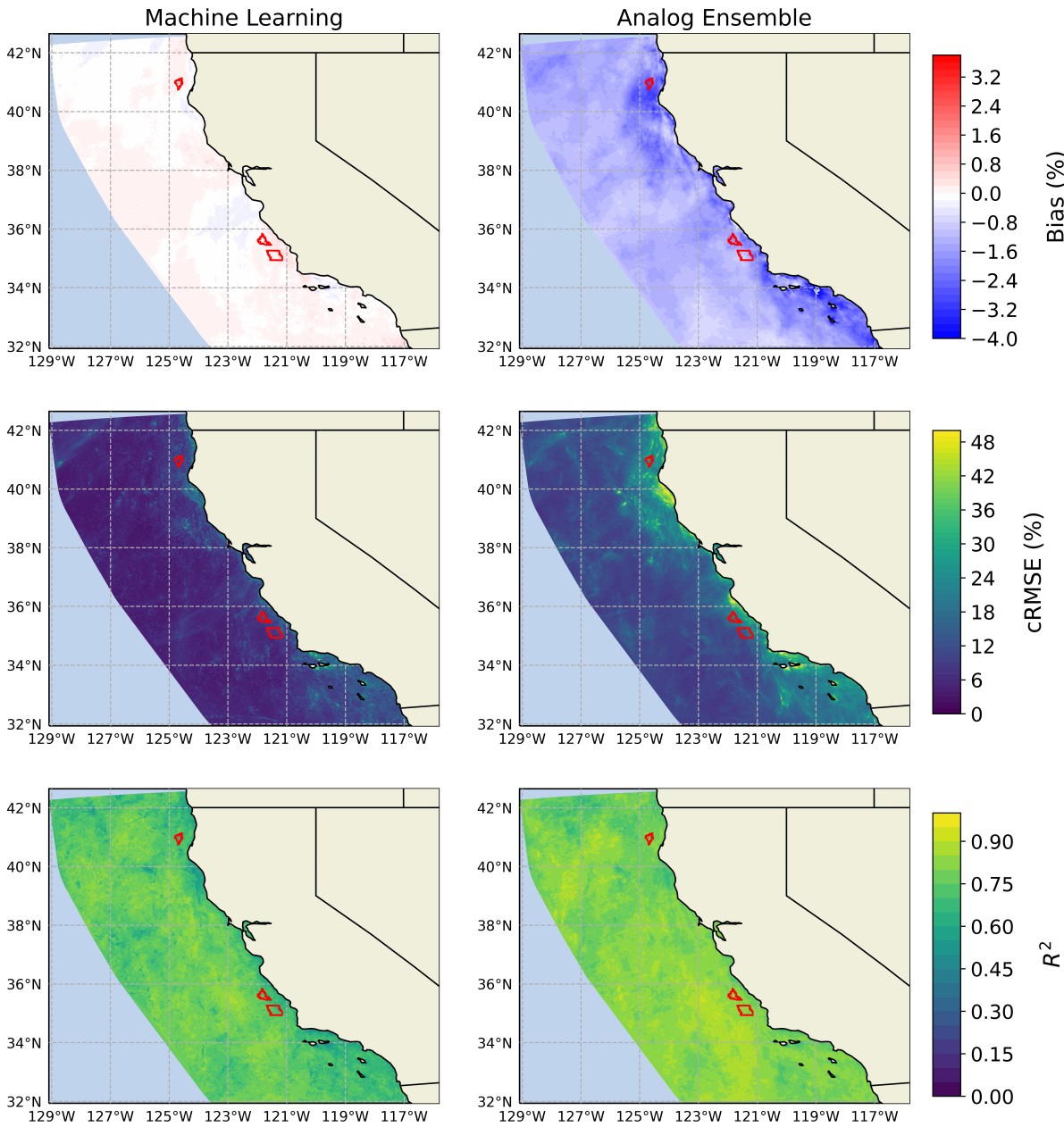

**Figure 4.** Map of testing bias, cRMSE, and $R^2$ determination coefficient from the GBM model (left) and the AnEn approach (right). The wind energy lease areas are highlighted in red.

240 (e.g., particularly high-wind-speed cases) when looking for similar atmospheric conditions in such a short repository. Also, when searching for the optimal number of analogs to use, there is always a trade-off between the prediction accuracy (e.g., the RMSE) and the prediction bias. For our analysis, the main goal was to maximize the prediction accuracy, in alignment with

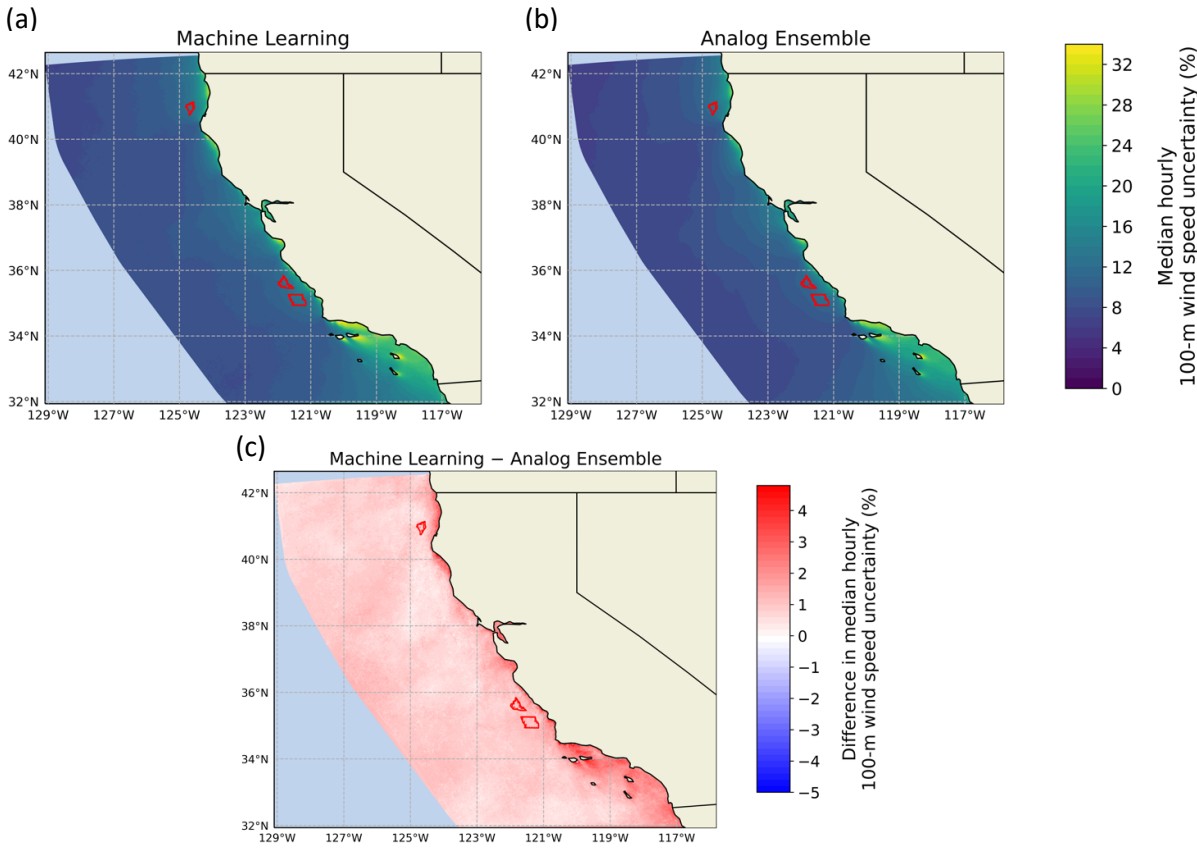

**Figure 5.** Median hourly boundary condition and parametric uncertainty for the 100-m wind speed, as derived from the machine-learning approach (a), the analog ensemble (b), and the difference between the two (c).

the ML approach, and therefore we set the RMSE as the optimization metric. During our grid search analysis to determine the optimal number of analog members, we observed that AnEn archives better bias with fewer members, which would however worsen the prediction accuracy (RMSE). Applying the bias correction proposed in Alessandrini et al. (2019), using a machine-learning similarity for analog definition (Hu et al., 2021b), or adopting a quantile mapping that uses quantiles of the analog ensemble instead of its mean (Sidel et al., 2020) would help reducing the AnEn bias, at the potential expense of computational costs.

### 3.2 Analysis of extrapolated wind speed boundary condition and parametric uncertainty

Now that the accuracy of both the proposed approaches has been assessed, we can analyze their long-term results. Figure 5 shows maps of the long-term median hourly boundary condition and parametric uncertainty for the 100-m wind speed predicted by the two proposed techniques, as well as the difference between the two. A strong agreement between the two approaches clearly emerges, with the AnEn approach predicting slightly lower values, as discussed from the analysis of the mean bias in

Figure 4. In general, we find a larger uncertainty close to land, with values locally greater than 30% of the mean wind speed, whereas in open waters the median hourly uncertainty is smaller than 10% of the WRF-predicted wind speed. The difference between the median prediction from the two approaches also gets larger close to the land. For the current three wind energy lease areas in the region, the machine-learning approach quantifies a long-term median hourly uncertainty between 12% and 14% at all three sites. On the other hand, the AnEn approach provides slightly lower values, between 10% and 13%, again with little variability across the three sites.

When focusing on offshore wind energy development, additional considerations are needed to understand how modeled wind speed uncertainty varies for the most relevant scenarios for energy production. When segregating data, having a long-term record allows for robust assessments of the variability among the considered classifications, which might otherwise have been much murkier when considering data from a short-term period only. Therefore, the twenty-fold increase of the size of the uncertainty data set provided by our proposed approaches brings an essential advantage to this direction.

Seasonality has a primary importance for the energy market, especially in a region such as California, with a strong peak in annual demand in summer, which recently led to detrimental rolling blackouts in the region. In this fragile scenario, assessing the uncertainty in the naturally varying long-term wind speed predictions could help assess the value that offshore wind energy can deliver to the California energy market, and achieve more accurate planning of the balance between supply and demand. Figure 6 compares maps of the seasonal deviation in median hourly normalized uncertainty for the 100-m wind speed in winter (December, January, February) and summer (June, July and August). For each season, the values shown are the difference between the median hourly uncertainty for that specific season and the overall median value (i.e., what is shown in Figure 5). For most of the considered domain, we find a larger sensitivity in WRF-predicted wind speed in the winter months, with the GBM showing a slightly larger seasonal deviation than the AnEn approach. At the Morro Bay and Diablo Canyon lease areas, the median winter uncertainty is between 2% and 8% larger than the annual median at the same locations. On the other hand, the Humboldt lease area shows a near-zero winter deviation, with the machine-learning approach predicting slightly increased winter uncertainty values, and AnEn predicting slightly negative ones. We find opposite results when considering the more energy-demanding summer months. Both Morro Bay and Diablo Canyon show a lower boundary condition and parametric uncertainty in summer, with a difference from their annual median values smaller than 4%. On the other hand, negligible variability is observed at the Humboldt lease area. We note that spring and fall months displayed intermediate results when compared to summer and winter (figures not shown).

Finally, we quantify the impact of different stability regimes on the long-term wind speed uncertainty. Various approaches to classify atmospheric stability offshore have been proposed and applied offshore (e.g., Archer et al. (2016)), including the shape of the wind speed profile, the use of the Richardson number, and of turbulent kinetic energy. Here, we classify atmospheric stability based on the bulk Richardson number, $Ri_B$, calculated over the lowest 200 m, as done in Rybchuk et al. (2021) for the same data set:

$$Ri_B = \frac{g\, z_{200}\, (\theta_{200} - \theta_0)}{0.5\, (\theta_{200} + \theta_0)\, \mathrm{WS}_{200}^2} \tag{3}$$

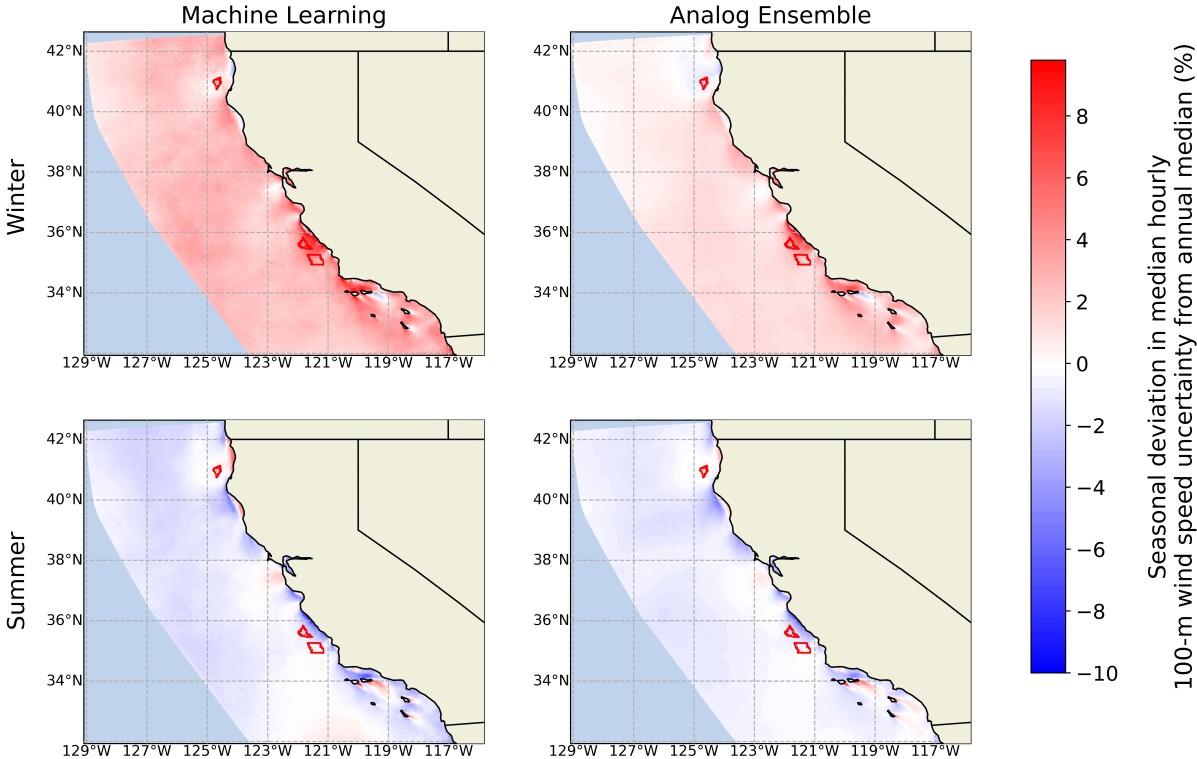

**Figure 6.** Seasonal deviation in median hourly normalized uncertainty in 100-m wind speed for winter (December, January, February) and summer (June, July and August), as derived from the machine learning approach (left) and the Analog Ensemble (right).

where $g = 9.81\,\mathrm{m\,s^{-2}}$ is the acceleration caused by gravity, $z_{200} = 200\,\mathrm{m}$, $\theta$ is potential temperature (K), and $\mathrm{WS}_{200}$ is the 200-m wind speed ($\mathrm{m\,s^{-1}}$). We consider stable conditions for $Ri_B > 0.025$, unstable conditions for $Ri_B < -0.025$, and near-neutral conditions otherwise. Figure 7 shows a histogram of the diurnal variability of the three stability regimes at the Humboldt wind energy lease area. We see a predominance of near-neutral and stable conditions, with a very weak diurnal variability. This is consistent with the sea surface temperature being generally colder than the near-surface air (because of ocean upwelling), which causes a predominantly stable stratification. Similar conditions are found at the other two wind energy lease areas. The maps in Figure 8 quantify how wind speed boundary condition and parametric uncertainty varies as a function of atmospheric stability. For each atmospheric stability class, we show the difference between the median hourly uncertainty for that specific stability condition and the overall median value (i.e., what is shown in Figure 5). The proposed approaches show a remarkable agreement. Neutral conditions show the lowest boundary condition and parametric uncertainty. At the three wind energy lease areas, we find uncertainty values about 2%-4% lower than the overall median in near-neutral conditions. On the other hand, the rare unstable cases show the largest uncertainty, with deviations up to +10% from the median at the considered wind energy lease areas. Finally, stable conditions also show positive deviations in uncertainty throughout the considered domain, with differences on the order of +2-5% at the wind energy lease areas.

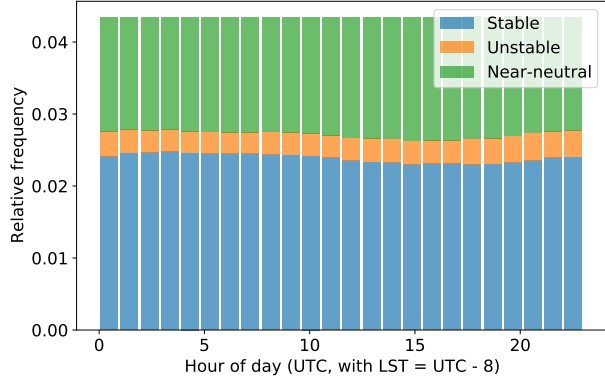

**Figure 7.** Daily distribution of atmospheric stability at the Humboldt wind energy lease area, as determined from the bulk Richardson number calculated over the lowest 200 m from the 20-year WRF simulation.

## 4   Conclusions

As offshore wind energy becomes a widespread source of clean energy worldwide, the importance of having an accurate, long-term characterization of the offshore wind resource is crucial, not only in terms of its mean value, but also of the uncertainty associated with this estimate. In our analysis, we focused on the California Outer Continental Shelf (OCS), where a significant

305   offshore wind energy development is expected in the near future, to propose innovative techniques to temporally extrapolate hub-height wind speed boundary condition and parametric uncertainty from a short-term mesoscale numerical ensemble to a long-term single model run. First, we propose a gradient-boosting model algorithm, in which a regression model is trained over the short-term numerical ensemble to predict its variability and then applied to the long-term single model run. We compare this technique with an analog ensemble (AnEn) approach, wherein the extrapolated uncertainty for each time stamp in the

310   long-term run is calculated by looking for similar atmospheric conditions within the short-term mesoscale numerical model ensemble. Adopting our proposed approaches for uncertainty extrapolation helps save significant computational resources as the desired long-term boundary condition and parametric uncertainty information can be derived from a much simpler setup wherein the computationally expensive numerical ensembles are only run over a short-term period.

   We find that both our proposed approaches agree well with the mesoscale model ensemble variability, thus providing a robust

315   representation of the long-term wind speed boundary condition and parametric uncertainty. While AnEn has a slightly larger $R^2$ coefficient with the mesoscale model across-ensemble data, we find that the gradient-boosting model has lower bias and centered root-mean-square error. However, we expect the AnEn performance to improve if either the bias correction for rare events proposed in Alessandrini et al. (2019) or the quantile mapping approach presented in (Sidel et al., 2020) is incorporated in the analysis. In general, we find that the offshore wind speed boundary condition and parametric uncertainty increases near

320   the coast. While the accuracy of the AnEn approach significantly degrades near the coast, the larger values in hub-height wind speed boundary condition and parametric uncertainty near the coast were also seen from the variability among the WRF ensembles (Optis et al., 2020), and attributed to diverging wind profiles associated with the choice of PBL scheme under strong

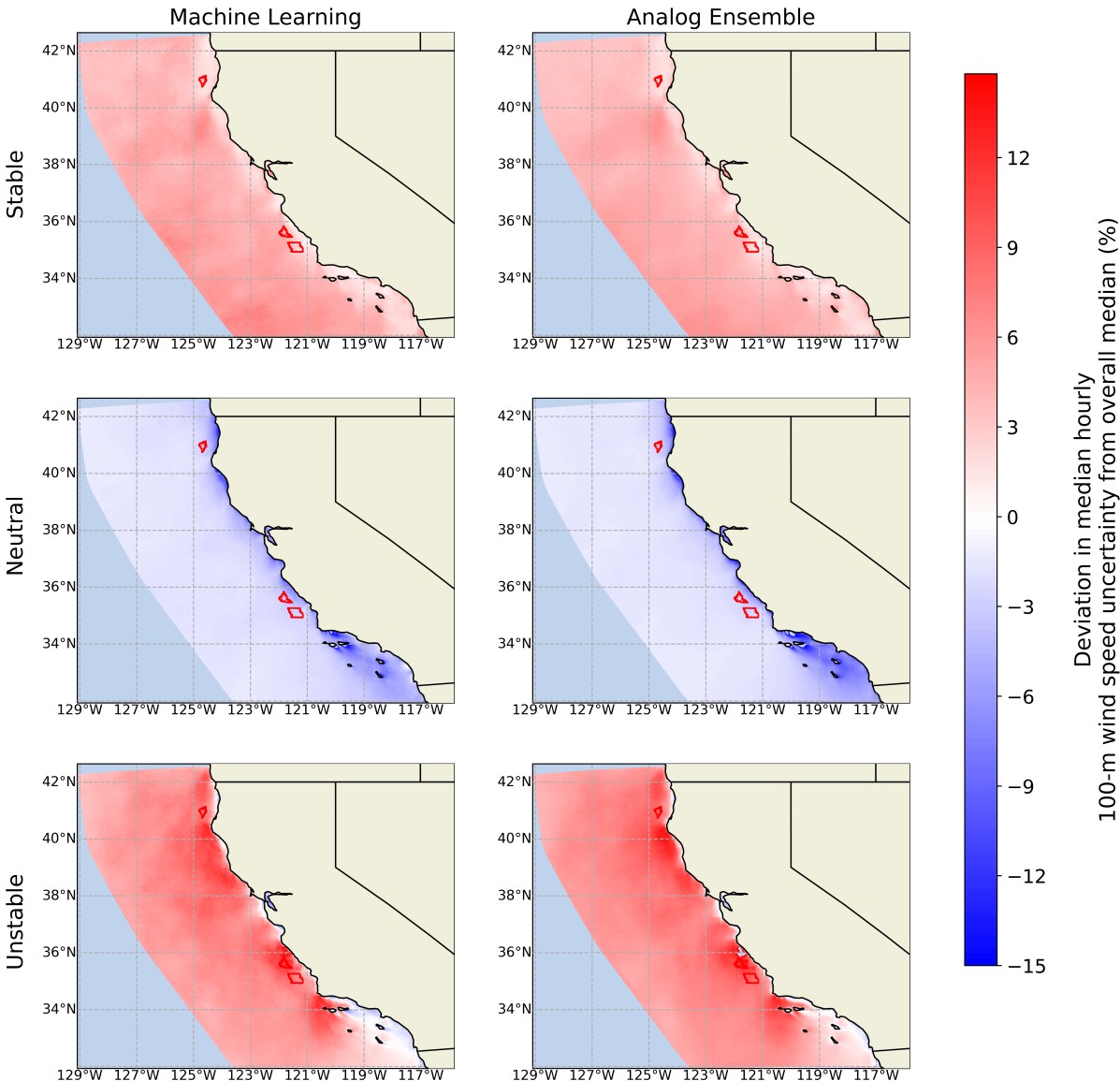

**Figure 8.** Deviation in median hourly normalized uncertainty for the 100-m wind speed from the annual median, for different atmospheric stability regimes, as derived from the machine-learning approach (left) and the AnEn approach (right).

stable atmospheric conditions near the coastline. We also find that uncertainty is larger in stable and unstable conditions, lower in near-neutral cases. On average, the hourly uncertainty at the current three wind energy lease areas in the California OCS is between 10% and 14% of their mean hub-height wind speed. Summer months also experience lower uncertainty, which will benefit the energy planning in a season with a strong demand, which has, in the past, led to detrimental rolling blackouts.

325

Clearly, the magnitude of the boundary condition and parametric uncertainty component that we quantified in our analysis is strictly connected to the (limited) number of choices sampled within the considered model setups. Given this underdispersive behavior of the numerical weather prediction ensembles (Buizza et al., 2008; Alessandrini et al., 2013), we expect the uncertainty quantified from our ensemble to be lower than the model error with respect to measurements. Still, we note that the same caveat would apply if the uncertainty was directly quantified by running a long-term numerical ensemble, and thus the computational advantages of our proposed approaches still hold. Moreover, we emphasize how the choices made to build our numerical ensemble represent either state-of-the-art resources or the most widely accepted choices within the wind energy modeling community. Also, a quantification and temporal extrapolation of the full uncertainty in modeled wind speed would require concurrent observations (and the knowledge of the inherent uncertainty associated with them) to be computed. Given all these considerations, many opportunities exist to further expand our work. While floating lidars with publicly available data have only been deployed in the California OCS very recently (Gorton, 2020), a few lidars have been deployed off the U.S. Eastern Seaboard for more than 1 year. Observations from long-term offshore meteorological towers are also available in the North Sea in Europe. Our analysis could be expanded by first comparing the model-related boundary condition and parametric uncertainty with the full modeled wind speed uncertainty calculated by comparing modeled data and observations. Then, our proposed approaches could be expanded to temporally extrapolate the full modeled wind speed uncertainty; for example, quantified in terms of the variability of the residuals between modeled and observed wind. Testing additional input features to the algorithms could also help further improving the accuracy of the proposed extrapolation. Also, the site specificity of the proposed approaches would need to be investigated to understand if a learning model trained at a site e.g., one ocean basin) can still provide accurate predictions when applied at a different location. Analog-based techniques could also be applied onshore, where the impact of more complex topography would likely need to be taken into account and incorporated in the algorithms. Finally, future work could focus on how interannual wind speed variability caused by climate change or long-term climatic and atmospheric oscillations (e.g., the North Atlantic oscillation) compares with the quantified uncertainty in modeled wind speed and how that should be taken into account for wind energy development purposes. To facilitate this extension, we have included in the Supplementary Materials a map of the interannual variability in 100-m wind speed quantified from the 20-year WRF run.

*Code and data availability.* Data from the WRF simulations over the California OCS are available at https://developer.nrel.gov/docs/wind/wind-toolkit/offshore-ca-download/. The code for the considered machine-learning model is available at https://github.com/nbodini/ML_UQ_offshore. The code for the AnEn approach is available at https://weiming-hu.github.io/AnalogsEnsemble.

*Author contributions.* NB and MO envisioned the analysis. MO ran the numerical simulation. NB performed the machine-learning analysis, in close consultation with MO. WH performed the AnEn analysis, with the guidance of GC and SA. NB wrote the majority of the manuscript, with significant contributions and feedback from all coauthors.

*Competing interests.* The authors declare that they have no conflicts of interest.

*Acknowledgements.* This work was supported and funded by the Bureau of Ocean Energy Management (BOEM) under Agreement No. IAG-360 19-2123 and by the National Offshore Wind Research and Development Consortium under Agreement No. CRD-19-16351. This research was performed using computational resources sponsored by the U.S. Department of Energy's Office of Energy Efficiency and Renewable Energy and located at the National Renewable Energy Laboratory. The authors thank Michael Rossol for his help in performing some of the computations using the National Renewable Energy Laboratory's High-Performance Computing Center.

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
