# Peer review of "Assessing Boundary Condition and Parametric Uncertainty in Numerical-Weather-Prediction-Modeled, Long-Term Offshore Wind Speed Through Machine Learning and Analog Ensemble"

_Wind Energy Science, 2021_

## Author Comment (AC1)

*In this document, the reviewer's comments are in black, the authors' responses are in red.*

We thank the reviewer for their thoughtful comments, which gave us an opportunity to revisit our analysis.

The manuscript describes a statistical (machine learning) approach to derive the uncertainty of wind data from mesoscale model simulations for longer time periods. The topic is relevant for wind energy. Thus, the manuscript should be published after revisions.

The manuscript is clearly written. Nevertheless, reading the manuscript rises a number of issues and questions which should be addressed in more detail in the manuscript.

(1) Lines 37-39: Using marine tower data (e.g., available offshore off the coasts of the Netherlands, Germany or Denmark) should at least be mentioned in the introduction in parallel to the possibility of using floating lidar data. There is more offshore wind data than those from buoys. Please add some references.

We have rephrased this part as: "Direct observations of the wind resource offshore are oftentimes limited to buoys, which offer measurements at very limited heights. Hub-height measurement of the wind resource offshore can be achieved with either offshore meteorological towers (e.g., Neumann et al. (2004); Fabre et al. (2014); Peña et al. (2014); Kirincich (2020)) or floating lidars (Carbon Trust Offshore Wind Accelerator, 2018; OceanTech Services/DNV GL, 2020). However, the often prohibitive costs connected to both these measurement solutions limit their availability to a handful of locations, despite recent efforts in leveraging their punctual hub-height measurements for wind speed vertical extrapolation over a larger region (Bodini and Optis, 2020; Optis et al., 2021a)."

(2) Given that there is some tower data (some of them for more than ten years), could testing of the results from this study against long-term time series from the FINO1 or FINO3 platform in the German Bight be an option?

Absolutely, using long-term tower data could be used in future work to model the actual error between WRF-modeled and observed wind speeds at hub-height. We had already mentioned this opportunity in the conclusions, but only with reference to floating lidar observations. We have now added an additional sentence to include tower data, too. That part now reads: "While floating lidars with publicly available data have only been deployed in the California OCS very recently (Gorton, 2020), a few lidars have been deployed off the U.S. Eastern Seaboard for more than 1 year. Observations from long-term offshore meteorological towers are also available in the North Sea in Europe. Our analysis could be expanded by first comparing the model-related boundary condition and parametric uncertainty with the full modeled wind speed uncertainty calculated by comparing modeled data and observations. Then, our proposed approaches could be expanded to temporally extrapolate the full modeled wind speed uncertainty; for example, quantified in terms of the variability of the residuals between modeled and observed wind."

(3) The addressed wind conditions in this manuscript are those in coastal areas. Therefore, the reviewer is wondering why coastal effects in the wind fields are not addressed. This should at least be mentioned in the Introduction.

We have added the following part to the Introduction: "Given these constraints, numerical weather prediction (NWP) models at the mesoscale are often used to obtain an in space and time continuous mapping of the available offshore wind resource at the heights relevant for commercial wind power plant deployment (e.g., Mattar and Borvarán (2016); Salvação and Soares (2018)), with some studies (Papanastasiou et al., 2010; Steele et al., 2013; Arrillaga et al., 2016) also focusing on the validation of modeled coastal wind effects, such as sea breezes, which have a significant impact on offshore wind energy production (Archer et al., 2014). "

(4) Some configurations of the mesoscale model WRF have problems representing the atmospheric boundary layer in the transition region between land to open sea (see, e.g., Siedersleben et al., 2018, DOI 10.1127/metz/2018/0900). Has the WRF configuration chosen for this study been checked for the ability to properly simulate the transition from land to sea?
We did not specifically check this, but wind flow in the region is almost entirely from the northwest, with large open fetch and minimal land-sea interaction. Therefore, we expect any impacts from improperly modeling the transition layer would be minimal in this context.

(5) This study addresses uncertainties in wind speed. How does the magnitude of these possible uncertainties in wind speed modelling relate to, e.g., interannual wind speed changes due to long-term oscillations in the global circulation (NAO etc.) or due to climate change?
Assessing the impact of climate change and/or long-term oscillations is strictly outside the scope of our manuscript. However, to guide the interested reader in this analysis, we have calculated the interannual variability in modeled 100-m wind speed from the 20-year record used in the analysis, and included the following map in the Supplementary Materials:

[Figure]

Also, we have added the following sentence at the end of the Conclusions:
"Finally, future work could focus on how interannual wind speed variability caused by climate change or long-term climatic and atmospheric oscillations (e.g., the North Atlantic oscillation) compares with the quantified uncertainty in modelled wind speed and how that should be taken

into account for wind energy development purposes. To facilitate this extension, we have included in the Supplementary Materials a map of the interannual variability in 100-m wind speed quantified from the 20-year WRF run."

(6) Figure 7: Why do unstable thermal stratification conditions have such a small share of all cases? Shouldn't the lagged annual SST variation (due to the much larger thermal inertia of the water) compared to the air temperature variation lead to roughly equally frequent stable and unstable situations (stable in late winter and spring, unstable in late summer and autumn)? Please give an explanation for the found bulk Richardson number distribution.
To justify the distribution of bulk Richardson number, we have calculated the difference between air temperature and sea surface temperature at the considered site, using the 20-year WRF data:

[Figure]

As the histogram shows, the air temperature is generally higher than the sea surface temperature, likely because of ocean upwelling in the region, which leads to a generally cold sea water. This induces a stable stratification, which justifies the distribution of bulk Richardson number found and reported in the paper. We have added a comment on this in the paper, too ("This is consistent with the sea surface temperature being generally colder than the near-surface air (because of ocean upwelling), which causes a predominantly stable stratification.").

(7) Why is the 2 m inverse Obukhov length chosen as a parameter at all in Section 2.2? This parameter can only be relevant for the whole rotor area of the future wind turbines in cases with perfect vertical mixing. In stable conditions the depth of the Prandtl layer is much shallower than turbine hub height so that the near-surface Obukhov length is most probably irrelevant for the 100 m wind speed. This is likely to be also true for cases in which internal boundary layers are present (which frequently happens in coastal areas).
See our answer to comment #9.

(8) Figure 3 demonstrates that the 2 m inverse Obukhov length is not a meaningful parameter, because it is always close to zero. Logically, the weight 0 has been attributed to this parameter in Table 4. So, once again, why has this parameter been included into this study?
See our answer to comment #9.

(9) Assumed that the inverse Obukhov length has nevertheless some relevance, a not-discussed contradiction is that there seems to be a slight bias of the inverse Obukhov length towards instability in Figure 3 while there is a strong bias of the bulk Richardson number towards stable conditions in Figure 7? This should be explained. It seems to confirm the irrelevance of the inverse Obukhov length stated above.
The observed range of inverse(L) is actually a quite typical one, if we consider that a value of L = +- 200m (a somewhat typical threshold between neutral and stable/unstable conditions) would translate in a 1/L threshold of +-0.005 m⁻¹. We have changed the x-axis limits in figure 3 to make the plot (where no real bias emerges) easier to understand.

That being said, we have performed the feature importance calculation for the input parameters to the ML algorithm at one test site (one of the three wind energy lease areas), and actually found how inverse(L) was the second most important input feature for the algorithm:

[Figure]

In any case, we agree with the reviewer that different choices for the stability parameter are possible, and using a different one could potentially improve the performance of the proposed algorithms (as hinted by the fact that inverse(L) was not deemed useful by the AnEn algorithm). We have added the following comment to Section 2.2: "We also acknowledge how different choices for the atmospheric stability parameter could be explored, potentially leading to a more accurate representation of stability at the heights of interest for wind energy development compared to the near-surface Obukhov length."

(10) A major finding of this study is the increase in wind speed uncertainty close to the coast and for non-neutral stratifications. Does this point to weaknesses in the used method or to weaknesses

in the used mesoscale model? Or is this a natural feature which comes out of this study? Please discuss this point in the Conclusions.

The larger uncertainty in modeled hub-height wind speed near the coast was also found in Optis et al. 2020 when looking at the WRF ensemble members only. In their report, they comment that "Most of this sensitivity can be attributed to the choice of PBL scheme (MYNN or YSU) and the diverging wind profiles associated with each under strong stable atmospheric conditions near the coastline." We have added the following sentences in the Conclusions of our paper: "In general, we find that the offshore wind speed boundary condition and parametric uncertainty increases near the coast. While the accuracy of the AnEn approach significantly degrades near the coast, the larger values in hub-height wind speed boundary condition and parametric uncertainty near the coast were also seen from the variability among the WRF ensembles (Optis et al. 2020), and attributed to diverging wind profiles associated with the choice of PBL scheme under strong stable atmospheric conditions near the coastline.".

---

## Author Comment (AC2)

*In this document, the reviewer's comments are in black, the authors' responses are in red.*

We thank the reviewer for their thoughtful comments, which gave us an opportunity to revisit our analysis.

The manuscript by Bodini et al. on "Assessing Boundary Condition and Parametric Uncertainty in Numerical-Weather-Prediction-Modelled, Long-Term Offshore Wind speed Through Machine Learning and Analog Ensemble" focusses on comparing two different methods on how to extrapolate the ensemble uncertainty of a short (one year time) series to a long time series for the wind conditions along the coast of California.

The study is generally well written, figures are selected meaningfully and readable. However, I have a number of minor comments to be taking into account before I can recommend publication in Wind Energy Science.

Minor Comments:

- Line 28: Northern Europe is typically referred to as Scandinavia. I guess you rather mean central Europe here?
We have rephrased this as "In some areas of the world, such as the North Sea in Europe,"

- Line 43: "a continuous, in space and time" → I suggest: an in space and time continuous
Changed.

- Line 76: "prohibitive, and innovative and more computationally" → remove the "and" before innovative
We hare rephrased this.

- Line 103: "The 15 WRF ensemble members" → I strongly recommend creating a table here with one row per one of the 16 members and then the settings in the columns. It is hard to read and understand here.
We had a typo in the sentence. Year 2017 (and not the 15 ensemble members, as previously stated in the draft) was selected because of strong data coverage from the network of buoy and coastal radar observations used for model validation. We think that now the section is clear: all 16 WRF ensembles share the common WRF attributes in Table 1; the list at page 5 shows the various combination of additional WRF parameters that led to the creation of the 16 ensemble members.

- Line 131: "mean hourly wind speed" → How did you compile the mean hourly wind speed? How many timesteps have you written out from the WRF runs to compute the average?
We have added the following: "(calculated from 5-minute WRF raw output)".

Line 147: "the WRF across-ensemble standard deviation" → what is the across-ensemble standard deviation. Is this the standard deviation from the ensemble members?
The WRF across-ensemble standard deviation is explicitly defined in equation 1.

Line 156-157: Can you explain why you used both the standard deviation from preceding 2 and 6 hours and how these time intervals were selected?

We selected these time intervals to try to capture variability of atmospheric conditions over different time scales which based on our experience could have an impact on hub-height wind speed uncertainty. As stated in our answer to the next comment, we have acknowledged in the paper that the choice of input features in our analysis was not exhaustive, and that "Testing additional input features to the algorithms could also help further improving the accuracy of the proposed extrapolation."

Line 161: "However, we found that including all the features" → can you explain how you found this? And did you try further features?
We compared the error metrics obtained (on a limited number of test sites) when using different sub-sets of input features, and found that the best results were obtained with all inputs together. We did not try additional features, but we agree that this might be interesting future work, and we have added the following sentence to the Conclusions: "Testing additional input features to the algorithms could also help further improving the accuracy of the proposed extrapolation."

Table 2: It is quite difficult to read column two. Maybe it helps to add a small empty row after each row?
Done.

Table 4: I think you should at least try to explain why there is no weight on the inverse Obukhov length but a strong weight on shear.
As suggested by the other reviewer, in stable conditions the depth of the Prandtl layer is much shallower than turbine hub height so that the near-surface Obukhov length is likely irrelevant for determining uncertainty in the 100 m wind speed. We have now added some comments on this in the paper.

Line 230 and Line 303-304: "Applying the bias correction proposed in" → So, if this would likely reduced the AnEn bias, why didn't you do it? This should at least be explained.
The negative bias is likely caused by two reasons, the limited historical repository to search and the fact we take an *average* of the 16 ensemble members. When only having a limited historical repository, there is a balance between the prediction accuracy and the systematic bias. During our grid search analysis for the best number of ensemble members to generate, we observed that AnEn archives better bias (closer to zero) with fewer members but the prediction accuracy, e.g., RMSE, would drop. To the focus of this manuscript, prediction accuracy is the most important metric, and therefore, we set RMSE optimization to be the most important metric. Additional bias correction processes might be able to help AnEn prediction, but we would like to keep a fair comparison between ML and AnEn. In fact, the postprocessing technique can be computationally expensive when applied to a large domain. Therefore, extensive correction on AnEn has not been explored. We have added information on this in the following paragraph, where we have also added reference to another technique that can be explored in future work to reduce the AnEn bias:
"The bias from the AnEn approach is likely because of the reduced length of the search period (1 year), which might be too limited for identifying a significant number (16) of analogs. This setup constrains the AnEn ability to account for rare events (e.g., particularly high-wind-speed cases) when looking for similar atmospheric conditions in such a short repository. Also, when searching for the optimal number of analogs to use, there is always a trade-off between the prediction accuracy (e.g., the RMSE) and the prediction bias. For our analysis, the main goal was to maximize

the prediction accuracy, in alignment with the ML approach, and therefore we set the RMSE as the optimization metric. During our grid search analysis to determine the optimal number of analog members, we observed that AnEn archives better bias with fewer members, which would however worsen the prediction accuracy (RMSE). Applying the bias correction proposed in Alessandrini et al. (2019), using a machine-learning similarity for analog definition (Hu et al., 2021b), or adopting a quantile mapping that uses quantiles of the analog ensemble instead of its mean (Sidel et al., 2020) would help reducing the AnEn bias, at the potential expense of computational costs."

Figures 6 and 8: There should be a space between caption text and unit.
Thank you for catching this, we have fixed this.

Line 279-280: "For each atmospheric stability class, the values shown are the" → Maybe an active sentence is better here, e.g., for each atmospheric stability class, we show....
Changed.

Line 293 and the whole Conclusions section: I suggest replacing Weather Research and Forecasting or the abbreviation WRF by "mesoscale" everywhere in the conclusion, because the results and conclusions should in principle be valid to any mesoscale ensemble dataset, shouldn't they?
We agree with the reviewer and have changed this as suggested.

Line 319: "deployed in the California OCS very recently" → I guess you mean this buoy? https://a2e.energy.gov/data/buoy/lidar.z06.00 Maybe it's worth referencing the dataset (including the doi) here.
We have added a reference as suggested.

References: The references are vastly incomplete. Every journal paper should have a doi and every technical report a link or other accessibility information.
We have updated the references whenever possible.